# Upconversion-Powered Photoelectrochemical Bioanalysis for DNA Sensing

**DOI:** 10.3390/s24030773

**Published:** 2024-01-24

**Authors:** Hong Liu, Weiwei Wei, Jiajun Song, Jin Hu, Zhezhe Wang, Peng Lin

**Affiliations:** 1Shenzhen Key Laboratory of Special Functional Materials & Guangdong Research Center for Interfacial Engineering of Functional Materials, College of Materials Science and Engineering, Shenzhen University, Shenzhen 518060, China; 2161120213@email.szu.edu.cn (H.L.); 2140120408@email.szu.edu.cn (J.S.); 2151120226@email.szu.edu.cn (J.H.); 2Fujian Provincial Key Laboratory of Quantum Manipulation and New Energy Materials, College of Physics and Energy, Fujian Normal University, Fuzhou 350117, China; zzwang@fjnu.edu.cn

**Keywords:** photoelectrochemical biosensors, upconversion nanoparticles, energy transfer, photo reabsorption

## Abstract

In this work, we report a new concept of upconversion-powered photoelectrochemical (PEC) bioanalysis. The proof-of-concept involves a PEC bionanosystem comprising a NaYF_4_:Yb,Tm@NaYF_4_ upconversion nanoparticles (UCNPs) reporter, which is confined by DNA hybridization on a CdS quantum dots (QDs)/indium tin oxide (ITO) photoelectrode. The CdS QD-modified ITO electrode was powered by upconversion absorption together with energy transfer effect through UCNPs for a stable photocurrent generation. By measuring the photocurrent change, the target DNA could be detected in a specific and sensitive way with a wide linear range from 10 pM to 1 μM and a low detection limit of 0.1 pM. This work exploited the use of UCNPs as signal reporters and realized upconversion-powered PEC bioanalysis. Given the diversity of UCNPs, we believe it will offer a new perspective for the development of advanced upconversion-powered PEC bioanalysis.

## 1. Introduction

Currently, advanced photoelectrochemical (PEC) bioanalysis is aggressively pursued among the electroanalytical community. PEC-based biosensing has a broad scope, encompassing various biomarkers such as nucleic acids, proteins, metabolites, cells, bacteria, and viruses [1,2,3,4,5]. The process of biomolecular recognition exhibits selectivity through mechanisms like DNA hybridization, immunological interactions, and enzymatic reactions. The detection of biomarkers enables the identification and quantification of various biological entities, facilitating research in molecular structure, disease detection and prevention, as well as ensuring food safety and monitoring environmental pollution [6,7,8]. To date, substantial efforts have recently been devoted to the design and development of photoelectrodes, signal transduction, and sensing modes for superior PEC detection [9,10,11,12]. In particular, innovative signaling transducers based on functional material are expected to be a powerful strategy to achieve advanced PEC bioanalysis [13]. Traditionally, ultraviolet or visible light is usually used as the illumination for photocurrent generation in PEC biosensors. Compared to ultraviolet or visible light, near-infrared (NIR) light offers advantages in bioanalysis for PEC biosensors. Its longer wavelength and lower energy enable deeper tissue penetration, minimal photobleaching, and reduced phototoxicity, making it a promising option for biosensing applications [14,15]. However, NIR-activated studies are seldom reported in the field of PEC bioanalysis, probably due to the fact that the low energy of NIR light cannot excite the electron–hole pairs of common semiconductors (>1.8 eV) [16,17]. Therefore, it is expected that photosensitive materials can be driven by low-energy NIR efficiently while achieving high efficiency PEC conversion for high-quality sensing platforms.

Upconversion nanoparticles (UCNPs) are a group of inorganic nanoparticles with the ability to exhibit anti-Stokes photoluminescence. Through a multi-photon absorption process, UCNPs can convert low-energy NIR light into higher-energy ultraviolet-visible (UV-vis) light. These UCNPs have been identified as promising photosensitive materials for various biological applications [18,19,20]. Due to their unique optical upconversion capability and outstanding properties such as deep penetrability, low light damage, and high light stability, UCNPs are gaining increasing interest in the construction of NIR-light-driven PEC biosensors [21,22,23,24,25]. For example, using NaYF_4_:Yb,Er particles combined with CdTe/TiO_2_ heterostructures, a NIR-light-motivated PEC interface was constructed for PEC aptasensing of cancer cells [16]. Very recently, NaYF_4_:Yb,Tm-based UCNP@SiO_2_@Ag@C-g-C_3_N_4_ core–shell nanospheres were developed for ultrasensitive PEC detection of *Escherichia coli* O157:H7 [26]. However, despite this progress, these exploitations are highly limited in the use of UCNP–semiconductor hybrids as photoelectrodes, in which a strong interfacial reflection between UCNPs and semiconductor has led to an unsatisfactory utilization of light sources [23,27,28]. To utilize light sources efficiently, we assume that UCNPs present a good reporter to directly transduce the biorecognition event via converting NIR light in PEC bioanalytical systems. Unfortunately, such a possibility has seldom been unveiled.

This study aims to investigate the concept of upconversion-powered PEC bioanalysis, which involves utilizing UCNPs as a reporter for efficient long-wavelength light harvesting. The UCNPs are employed to align their upconversion phenomena with wide bandgap photosensitive semiconductors on the photoelectrode. This alignment activates the semiconductors for photocurrent generation, and the system is evaluated using DNA recognition. For proof-of-concept experiments, CdS quantum dots (QDs) modified on indium tin oxide (ITO) were used to construct the photoelectrode. NaYF_4_:Yb,Tm@NaYF_4_ UCNPs (hereafter briefed as UCNPs) were served as the NIR-light motivated PEC interface, which was confined by the double-stranded DNA hybridization. The visible-light-absorption of CdS QDs spectrally overlaps with the emission of the UCNPs, which makes it possible to use the latter as luminescent material to convert NIR light and excite the former. Meanwhile, energy transfer occurred when the UCNPs were placed in close proximity to the CdS QDs. By measuring the photocurrent change, target single-stranded DNA was detected in a specific and sensitive way with a detection limit of 0.1 pM. This work presents the use of UCNPs as signal relay and the resulting upconversion-powered PEC bioanalysis. Given the diversity of UCNPs and the superior benefits of NIR light, we believe this presents a different perspective for the general development of advanced upconversion-powered PEC bioanalysis.

## 2. Materials and Methods

### 2.1. Materials and Reagents

Indium tin oxide (ITO) was purchased from Zhuhai Kaivo Optoelectronic Technology Co., Ltd., Zhuhai, China. Ascorbic acid (AA) and monoethanolamine (MEA) were obtained from Shanghai Aladdin Biochemical Technology Co., Ltd., Shanghai, China. Poly(diallyldimethylammonium chloride) (PDDA), 1-ethyl-3-(3-dimethylaminopropyl) carbodiimide (EDC), and N-hydroxysuccinimide (NHS) were obtained from Sigma-Aldrich Co Ltd., Shanghai, China. UCNPs were obtained from Hangzhou Fluo NanoTech Co., Ltd., Hangzhou, China. DNA sequences were obtained from Genscript Biotech Co., Ltd., Nanjing, China, and the corresponding sequences are summarized in Appendix A. All other reagents were of analytical grade and were used as received.

### 2.2. Designing and Assembling of the Photoelectrode

The ITO slices were cleaned by boiling in 2 M KOH solution dissolved in 2-propanol for 20 min, and then were washed by abundant amounts of DI water. CdS QDs were synthesized by using a water-based method and then immobilized onto the ITO electrode with an area of 0.3 cm^2^ using the previous technology [29]. Then, the ITO electrode modified with CdS QDs was immersed in a mixture of EDC (20 mg/mL) and NHS (10 mg/mL) for 1 h at room temperature. After that, 25 μL of 1 μM probe DNA was dropped onto ITO electrodes and kept for 12 h at 4 °C. The interaction between CdS QDs and DNA fragments is covalent bonding through carboxyl modified on CdS QDs and amino modified on DNA fragments. Then, the electrodes were carefully washed with 10 mM PBS (pH 7.4) solution to remove the non-immobilized DNA probes. Subsequently, the electrode was blocked by 1 mM MEA for 2 h at 4 °C to eliminate excess carboxyl groups.

### 2.3. Construction of UCNPs Reporter and Detection of Target DNA

Firstly, UCNPs were covalently connected with target DNA. Specifically, 50 μL of 10 μM UCNP aqueous solution and 100 μL of 10 mM EDC were added to 1 mL of 10 mM PBS and kept at room temperature for 30 min. Then, 50 μL of 10 mM NHS and 50 μL of 10 μM target DNA were added to the above solution and kept at room temperature for 3 h. The junctional complex was collected by centrifugal filtration and the sediment was redispersed by 10 mM PBS. Amine-contained target DNA was connected with UCNPs capped with carboxylic acid groups via the classic EDC/NHS coupling reaction. The oligonucleotide sequences of the 24 base pairs used here are listed in Appendix A. Unconnected target DNA and UCNPs were removed by using centrifugal filtration.

For target DNA detection, 25 μL of target DNA labeled with UCNPs was dropped onto the photoelectrodes and kept for 1 h at 37 °C to achieve the hybridization. Then, the electrodes were washed three times with 10 mM PBS (pH 7.4) to remove the unhybridized DNA targets.

### 2.4. Characterizations and Measurements

The photoluminescence (PL) spectra were obtained by using a 970 CRT Fluorescence Spectrophotometer (INESA, Shanghai, China). The UV-vis absorption spectra were obtained by using a UV-2450-visible Spectrophotometer (Shimadzu Corporation, Kyoto, Japan). The Fourier-transform infrared (FTIR) spectra were obtained by using a Nicolet-6700 Fourier Infrared Spectrometer (Thermo Fisher, Waltham, MA, USA).

The successful preparation of the UCNP-powered PEC biosensor was confirmed by using a regular PEC system with three electrodes, where a modified ITO electrode with an area of 0.3 cm^2^ was used as working electrode, a platinum plate electrode as the counter electrode, and a saturated Ag/AgCl as the reference electrode. The electrolyte was 0.1 M PBS containing 0.1 M AA, in which AA served as the electron donor. Photocurrent was measured by using a CHI 660E electrochemical workstation (Chenhua, Shanghai, China) at a constant potential of 0 V (vs. saturated Ag/AgCl). The excitation wavelength was 980 nm under the dark environment.

## 3. Results and Discussion

### 3.1. Characterization

The principle of the UCNP-powered PEC bioanalysis construct for DNA detection is schematically described in Figure 1. Carboxyl-functionalized CdS QDs were modified on ITO substrate and then covalently bound with amino-modified probe single-stranded DNA (ssDNA). On the other hand, an NaYF_4_:Yb,Tm core was coated with an undoped NaYF_4_ shell and then coated with a polyethylene glycol (PEG) layer for further carboxylation with amino-modified target ssDNA. Appendix A shows the FTIR spectrum of the UCNPs, which proves that carboxyl groups were successfully modified on the surface of the UCNPs. Hybridization of the probe ssDNA and target complementary ssDNA brings the distance between UCNPs and CdS QDs closer, to within 10 nm, as the DNA sequence is 24 base pairs (about 1 nm for 3 base pairs). UCNPs possess the ability to convert near-infrared excitation (980 nm) into visible emissions that align with the absorption spectral range of CdS QDs. Therefore, the fluorescence of luminescent UCNPs could serve as an excitation source to activate CdS QDs for photocurrent generation. The concentration of target DNA determines the intensity of fluorescence emitted by UCNPs, and then the level of photocurrent. By measuring the photocurrent change, the target DNA can be sensitively detected.

UCNPs and CdS QDs are the materials critical for the novel PEC biosensor. For successful construction of the bioanalytical system, the size and structure of UCNPs and CdS QDs were investigated. Figure 2a,b shows the transmission electron microscopy (TEM) image and high-resolution transmission electron microscopy (HRTEM) image of the NaYF_4_:Yb,Tm@NaYF_4_ sample. It can be seen that the NaYF_4_ shell was uniformly coated on the surface of the NaYF_4_:Yb,Tm core, indicating a relativity to a decrease in surface defects of the lattice and concomitantly the energy dissipation reduction, as well as to a light-harvesting improvement [30,31]. Moreover, a PEG-COOH layer was uniformly coated on UCNPs, with a thickness of around 3.5 nm. Figure 2c shows the TEM image of CdS QDs with an approximate diameter of 5 nm. A typical UV-vis absorption spectrum of the prepared CdS QDs and a photoluminescence (PL) spectrum of the UCNPs were obtained to prove the stable and effective upconversion luminescence to the excitation of CdS QDs. As shown in Figure 2d, two emission peaks around 450 nm and 475 nm were resolved for the core–shell nanoparticles (UCNPs), suggesting an excellent fluorescence intensity which is much higher than that of the core-only structure [32,33]. Meanwhile, there is a distinct absorption of light under 500 nm for the CdS QDs. The overlap of the two spectra ensures that the fluorescence of the UCNPs can effectively excite the CdS QDs to transfer the electrons in the valence band to the conduction band.

### 3.2. Sensitive Detection of DNA

After construction of the sensing platform, PEC behaviors for DNA detection were verified and analyzed. A typical time-based photocurrent response for each step measured in 0.1 M PBS containing 0.1 M AA was recorded under 980 nm illumination, as shown in Figure 3a–c. Specifically, at the first stage, the CdS QD-modified ITO electrode showed a weak NIR photocurrent under illumination (CdS curve of Figure 3a, black curve) because of the fractional absorption of near-infrared light by CdS QDs. Compared with the photocurrent generated by the UV-vis light-activated CdS QD-modified ITO electrode, the magnitude was lower by one or two orders [34,35]. After the probe DNA was grafted to CdS QDs, the photocurrent decreased a lot as depicted in ssDNA curve (red curve) when compared with the initial intensity in the CdS curve (black curve). This decrease could be mostly attributed to the steric hindrance effect, in which the probe ssDNA and MEA had a certain volume and would impede the electron transfer in PEC system. After that, the UCNP-powered PEC system was used for the detection of target ssDNA, in which the double-stranded DNA (dsDNA) was obtained by the hybridization of probe ssDNA and target complementary ssDNA. The platform was first exposed to the UCNP-labeled target DNA with a concentration of 10^−6^ M. We found that there was an apparent inverse increase in the photocurrent, as shown in dsDNA curve (blue curve). This increase in photocurrent was mainly due to the hybridization of target complementary DNA with probe DNA, which introduced the NIR-light-activated UCNPs into the system. During exposure to target DNA, UCNPs were placed in close proximity to CdS QDs. By adjusting the concentrations of Yb^3+^ and Tm^3+^ ions, the primary emission peaks of UCNPs were tuned to 450 nm and 475 nm when excited by NIR light at 980 nm. Simultaneously, the synthesized CdS QDs exhibited an efficient absorption of visible light below 500 nm. Consequently, the fluorescence emitted by UCNPs effectively served as an excitation source to activate the CdS QDs, thereby enabling the generation of photocurrent. Appendix A shows the photoluminescence (PL) spectrum and UV-vis absorption spectrum of the target DNA which connected with/without UCNPs. We can see that individual DNA shows no PL intensity around 450 nm and 475 nm, and individual UCNPs show no unique absorption peak at 260 nm, proving the efficient connection of UCNPs and target DNA. Therefore, the target DNA could be detected by measuring the photocurrent change. The UCNP-powered PEC biosensor for DNA detection provides a new method with covalently immobilized UCNPs as a reporter and QDs as an acceptor, which is different from other UCNP-driven PEC sensors with UCNP–semiconductor hybrids as photoelectrodes [16,22,23,26].

Using the same method, the detection limit of target DNA was studied by varying the concentration from 1 × 10^−13^ M to 1 × 10^−6^ M, while the concentration of probe DNA was fixed at 1 μM, as shown in Figure 3d. Under each condition, at least three samples were tested. Here, we use the percentage of photocurrent change (Δ*I*/*I*) to represent the signal response caused by the hybridization of target DNA with probe DNA, which is given by the following equation:(1)$$\Delta I/I=\frac{I_{\mathrm{dsDNA}}-I_{\mathrm{ssDNA}}}{I_{\mathrm{ssDNA}}}$$
where *I*_ssDNA_ and *I*_dsDNA_ correspond to the photocurrent response after the immobilization of probe DNA and the detection of target DNA, respectively. As shown in Figure 3d, Δ*I*/*I* increased accordingly with the increasing concentration of target DNA, suggesting a near-linear relationship from 1 × 10^−11^ M to 1 × 10^−6^ M. An increase of 30.2% in Δ*I*/*I* was found when the concentration of target DNA was 10^−6^ M, while an increase of only 4.7% was observed for the 10^−13^ M of target DNA. As the Δ*I*/*I* caused by pure PBS solution (baseline noise) was only 1.2%, the detection limit for target DNA by our UCNP-powered PEC-based biosensor was considered to be 10^−13^ M (0.1 pM). The reproducibility of the sensor is important and effective for verification of the test result and stability of the platform. Hence, six sets of assays were conducted under identical conditions, with a target DNA concentration of 1 μM. The photocurrents of the electrodes showed minimal variation at each step involving the modification of CdS QDs, ssDNA, and dsDNA (Appendix A). These results serve to validate the reproducibility and stability of the detection platform.

Actually, the steric hindrance effect caused by the hybridization of target DNA may hinder electron transfer in the PEC system. So, control experiments which measured the hybridization of target DNA without labeled UCNPs were conducted for comparison. As shown in Figure 4a, in the absence of UCNP activation, the photocurrent declined further after the hybridization of unlabeled target DNA in comparison to the photocurrent of the photoelectrode with probe DNA. The further decline in photocurrent was deduced by the increased space-hindered effect based on the unlabeled target DNA. As shown in Figure 4b, −Δ*I*/*I* decreased with the decrease in the concentration of unlabeled target DNA, suggesting a detection limit of only 10 nM (much worse than that of UCNP-powered sensors). The results clearly show that the observed increase in photocurrent in the UCNP-powered PEC bioanalysis was achieved by overcoming the steric hindrance effect induced by the target DNA itself. This highlights the significant impact of UCNPs in the system.

### 3.3. Energy Transfer of Upconversion-Powered Photoelectrochemical DNA Sensor

The sensing mechanism of the UCNP-powered PEC DNA sensor was studied. Figure 5 illustrates in detail the proposed electron and energy transfer route of the whole process, which refers to the excitation, upconversion, photo reabsorption (PR), energy transfer (ET), and emission. In the NaYF_4_:Yb,Tm structure, Yb^3+^ ions show a strong absorption of light at 900–1000 nm. Under the 980 nm illumination, Tm^3+^ ions are triggered to their high-energy levels through energy-transfer upconversion from excited Yb^3+^ ions to Tm^3+^ ions. Corresponding to the energy level transfer of ^1^D_2_ → ^3^F_4_ (450 nm) and ^1^G_4_ → ^3^H_6_ (475 nm), fluorescence levels of 450 nm and 475 nm are emitted. Then, on the one hand, the blue emissions of UCNPs are just in the absorption range of CdS QDs and can be absorbed by CdS QDs. The blue emission of UCNPs excites the transfer of CdS QDs electrons from the valence band to the conduction band and, subsequently, further transfer to the ITO electrode for the generation of photocurrent. Thus, it effectively avoids the radiative decay of CdS QDs. In this process, the 0.1 M ascorbic acid (AA) electrolyte served as an electron donor to provide enough electrons for electron transfer. On the other hand, the ^1^G_4_ energy level of the Tm^3+^ ion is higher than the conduction band of the CdS QDs, and there is a close distance within 10 nm between the UCNPs and CdS QDs, which is the length of DNA. As a result, an efficient energy transfer can occur between the two energy bands, leading to photocurrent generation caused by electron transfer to the ITO electrode. Some works have reported biosensors based on the energy transfer effect [36,37,38]. Powered by upconversion absorption together with the energy transfer effect through UCNPs, a photocurrent change can be detected for DNA sensing in the proposed PEC bioanalysis method.

### 3.4. Selectivity of DNA Sensor

In addition, the selectivity of the PEC-based biosensor was also investigated by evaluating the performance of the platform. Interference experiments were conducted with target complementary DNA (cDNA), two non-complementary DNA strands (ncDNA1 and ncDNA2), and background solution (0.1 M PBS, pH 7.4) as control experiments following the same measuring procedure. A sufficient amount of analyte with a concentration of 1 × 10^−6^ M was applied for better signal response, so as to verify the selectivity. As shown in Figure 6, no obvious photocurrent change occurred due to the non-specific effect of the two non-complementary DNA strands and background solution. Compared with the 30.2% photocurrent change for the target cDNA, the signal response changes in the other three non-specific analytes were all below 2.5%, which indicated the excellent specificity of the UCNP-powered PEC biosensor.

## 4. Conclusions

In summary, this study presents a novel UCNP-powered PEC bioanalysis construct for DNA sensing. In this system, NaYF_4_:Yb,Tm@NaYF_4_ UCNPs were confined onto an CdS QD-modified ITO electrode via DNA hybridization. UCNPs were employed as the NIR-light-driven PEC interface, where the CdS QD-modified ITO electrode utilized upconversion absorption and energy transfer from the UCNPs to generate photocurrent. This was facilitated by the spectral overlap and matching energy levels between UCNPs and CdS QDs. By measuring the photocurrent change, the target complementary DNA could be detected in a specific and sensitive way with a detection limit of 0.1 pM. This work proposes the use of UCNPs as signal relay and in upconversion-powered PEC bioanalysis. Given the diversity of UCNPs and the superior benefits of NIR light, we believe it will present a different perspective for the general development of advanced upconversion-powered PEC bioanalysis.

## Figures and Tables

**Figure 1 sensors-24-00773-f001:**
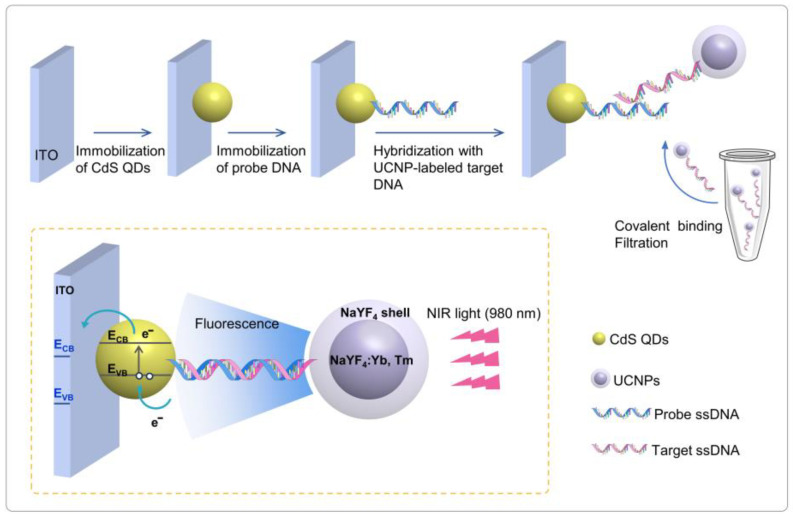
Schematic diagram of UCNP-powered PEC bioanalysis for DNA sensing.

**Figure 2 sensors-24-00773-f002:**
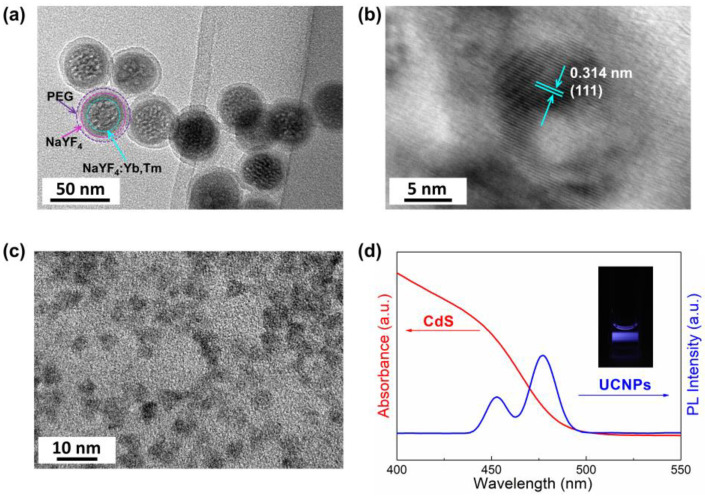
(**a**) TEM image of PEG-COOH coated NaYF_4_:Yb,Tm@NaYF_4_ UCNPs. (**b**) HRTEM image of NaYF_4_:Yb,Tm core, which is the cubic phase, where the interplanar spacing of (111) plane is 0.314 nm. (**c**) TEM image of CdS QDs. (**d**) The UV-vis absorption spectrum of prepared CdS QDs (red) and the photoluminescence (PL) spectrum of UCNPs (blue); the excitation wavelength is 980 nm. Inset is photograph of the UCNPs in water under 980 nm laser illumination.

**Figure 3 sensors-24-00773-f003:**
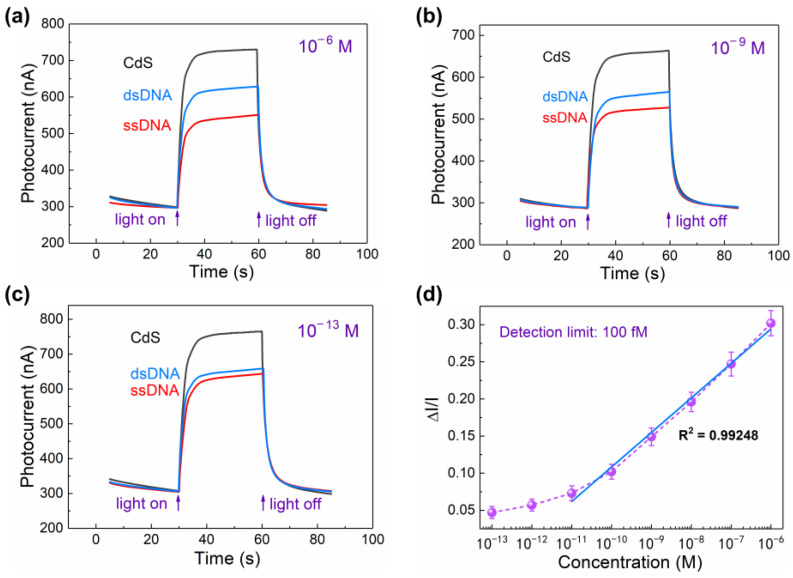
Photocurrent responses measured in 0.1 M AA solution after the modification of CdS QDs (black curve), subsequent modification with probe DNA (red curve), and hybridization with target DNA labeled with UCNPs (blue curve). The excitation light wavelength is 980 nm. The concentrations of target DNA are (**a**) 10^−6^ M, (**b**) 10^−9^ M, and (**c**) 10^−13^ M. (**d**) The percentage of photocurrent change Δ*I*/*I* after hybridization with target DNA at different concentrations.

**Figure 4 sensors-24-00773-f004:**
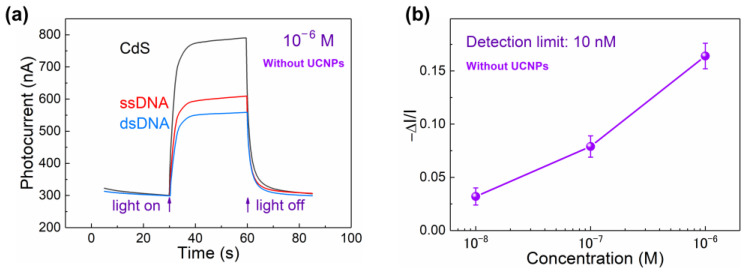
(**a**) Photocurrent responses measured in 0.1 M AA solution after modification of CdS QDs (black), probe DNA (red), and target DNA (blue) without labeled UCNPs. The excitation wavelength is 980 nm. The concentration of target DNA is 10^−6^ M. (**b**) The percentage of photocurrent change after reaction with different concentration of target DNA without labeled UCNPs.

**Figure 5 sensors-24-00773-f005:**
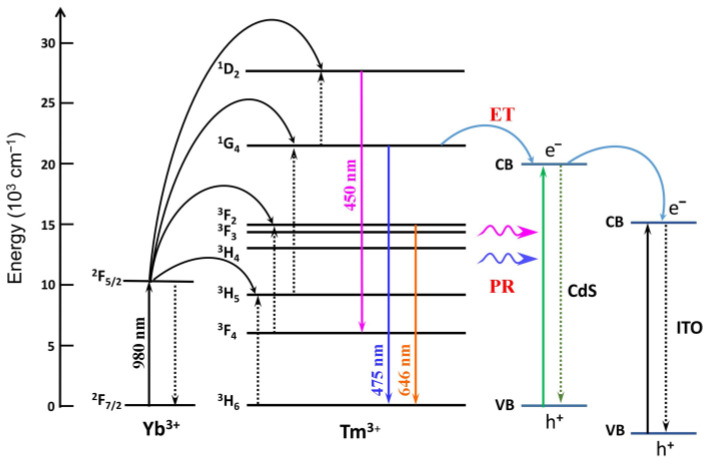
Schematic of the photo reabsorption (PR) and energy transfer (ET) between UCNPs reporter and CdS QDs acceptor. The upconversion emission peaks at 450 and 475 nm originated from ^1^D_2_ → ^3^F_4_ and ^1^G_4_ → ^3^H_6_ transitions of activator Tm^3+^, respectively.

**Figure 6 sensors-24-00773-f006:**
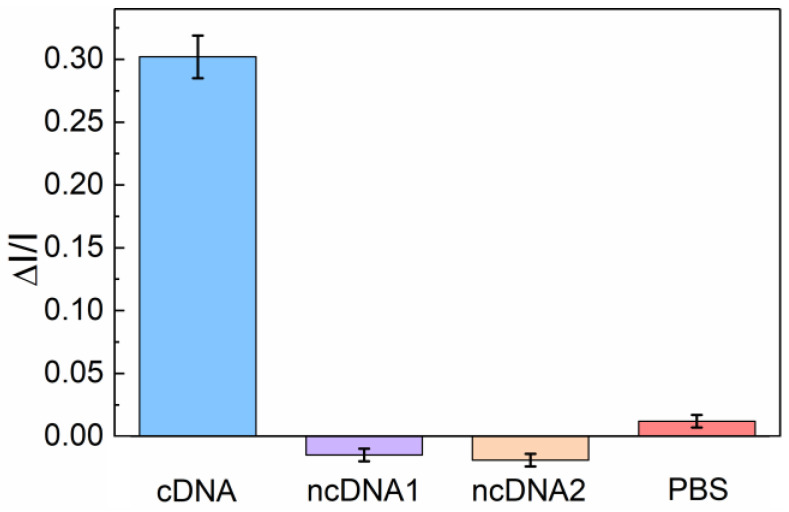
Selectivity of the UCNP-powered PEC bioanalysis: Δ*I*/*I* for the detection of target cDNA (10^−6^ M), two non-complementary ncDNA1 and ncDNA2 (10^−6^ M), and PBS solution (0.1 M, pH 7.4).

## Data Availability

Data are contained within the article and Appendix A.

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
