# Peer review of "Upconversion-Powered Photoelectrochemical Bioanalysis for DNA Sensing"

_sensors, 2024, doi:10.3390/s24030773_

Round 1

Reviewer 1 Report

Comments and Suggestions for Authors

Authors present a concept of upconversion-powered photoelectrochemical bioanalysis using UCNPs and QDs to detect DNA hybridization. They show that the energy transfer concept is functional and PEC analysis works for DNA sensing with a relatively large range of DNA concentration. The concept itself is interesting, but the application in which the functionality is shown, is not relevant as such. After improvements, the concept can be expected to be published, but some major points (listed below) needs to be considered before that.

1. Authors have shown the functionality using two surface conjugated complementary DNA. This method tells basically how much UCNP is needed to create a response but nothing about the functionality of the assay itself. System should be tested in competitive format having a single UCNP-oligo concentration and multiple concentrations of the competing oligo (can be the same as conjugated with UCNP). This would prove the correct functionality and show if the system is usable for ''true'' assay. At the moment, either of these have been shown in a reasonable enough way. ''Negative'' effect without DNA makes this factor even more crucial. 

2. Overall the writing style makes the manuscript difficult to read and in some cases difficult to understand. This is basically because of two thing; 1. extra long and difficult sentences and 2. writing style going picture by picture and not really telling the story so that pictures would support the text. For the first point, numerous examples can be given, for example P1 lines 30-33 and 39-42 or P2 lines 56-60. Picture by picture writing format goes through the whole manuscript and causes also lack in discussion at some point in the results section.

3. Manuscript is also slightly imbalanced as it basically shows sensor design, but without enough background references. On the other hand, it shows bio-related application which is not sufficient as such and also it is clear that the authors are not familiar to write bio-related texts or don't have information enough to write bio-related text. The work must be improve by bringing it more in the context by at least increasing the number of references.

4. SI looks now like a student work, and as there is no page limit, things can be more carefully explained in the figure legend. For example in S1 there could be comparative data making more sense to explain the spectra.

Comments on the Quality of English Language

The English language is mostly sufficient, but the structure of the text is not. Long and difficult sentences are problematic and also some part having repetition makes the reading hard. Additionally, figure by figure structure makes discussion too superficial.

Reviewer 2 Report

Comments and Suggestions for Authors

In this paper, a self-powered model for photoelectric bioanalysis is proposed. This idea is relatively new, but it has been reported as early as in Analytical Chemistry (AC) journal. After the current version requires some key data supplements and text refinements, it may be published in the sensor journal. My advice is to make major revision. The specific opinions are as follows:

1. The current title is too large (the current title is a bit like a review), please redefine the title according to the research content.

2. The expression order of 1uM to 10 pM in the abstract is wrong, which should be from small to large.

3. Please specify the size of ITO.

4. What is the interaction between CdS quantum dots and DNA fragments, please elaborate in the article? Why rotating CdS as the substrate material? Is it universal (such as ZnS).

5. Complementing the electrochemical characterization experiments (CV and EIS) during the interface repair process is critical for photoelectrochemical sensing.

6. The particle size distribution of Fig.2a and 2c is given, and whether Fig.2b can give the crystal plane is given.

7. How is the reproducibility of the sensor? What is the stability? For the basic characteristic parameters of the sensor, the author needs to supplement the experiment to explain.

8. Please check the reference format carefully and update some relevant appropriate references. (References: Microchemical Journal 174 (2022) 107038, etc.).

Comments on the Quality of English Language

Minor editing of English language required

Round 2

Reviewer 1 Report

Comments and Suggestions for Authors

Manuscript is significantly improved, and I understand that this is a prove of concept. However, using DNA detection limit type of terms is misleading as that is really not measured. Detection limit is for the particle and detection, and DNA is just a tool to do this.
